# Influence Functions for Preference Dataset Pruning

**Daniel Fein**
Stanford University
drfein@stanford.edu

**Gabriela Aranguiz-Dias**
Stanford University
gadias@stanford.edu

## Abstract

Language models are commonly fine-tuned via reinforcement learning to alter their behavior or elicit new capabilities. Datasets used for these purposes, and particularly human preference datasets, are often noisy. The relatively small size post-training datasets, combined with parameter-efficient fine-tuning methods, enable the use of influence functions approximations to detect and prune training examples that are harmful to performance on a validation set. In this work, we adapt the TL;DR dataset for reward model training to demonstrate how conjugate-gradient approximated influence functions can be used to filter datasets. In our experiments, influence function filtering yields a small retraining accuracy uplift of 1.5% after removing 10% of training examples. We also show that gradient similarity outperforms influence functions for detecting helpful training examples. This suggests that local curvature is important for detecting harmful training examples, but less so for identifying helpful examples.

## 1 Introduction

Influence functions are powerful tools for understanding how specific data affect the behavior of neural networks trained via gradient descent [Koh and Liang, 2017]. However, their practicality for language models is encumbered by their reliance on the inverse-hessian with respect to model parameters. The large scale of language models, both in terms of parameter count and training data, thus poses a challenge for influence-function based data attribution on the scale of pre-training. Despite this, approximations of the inverse hessian used at scale have demonstrated the power of influence functions to reveal insights into how language models make use of their training data Grosse et al. [2023].

Recently, the post-training paradigm has proven crucial for controlling the behavior of language models both with respect to helpfulness and harmlessness as well as math and coding abilities [Bai et al., 2022b, Zelikman et al., 2022]. Meanwhile, techniques for parameter-efficient fine-tuning such as Low-Rank Adaptation enable this post-training to be done via training of relatively few model parameters Hu et al. [2022]. The outsized effectiveness of this much less computationally-expensive training warrants a new investigation into influence functions for understanding the relationship between data and model behavior.

Turning specifically to the task of human preference modeling, it is estimated that 20-40% of training data used to align language models with human preferences is noisy [Gao et al., 2024]. Influence functions have been used to reduce the size of training sets for computer vision tasks [Yang et al., 2023], but have not yet been applied to language model data curation.

In this work, we demonstrate how influence functions approximated via the conjugate gradient method may help create stronger reward models via filtering out of noisy examples.

## 2 Related Works

**Influence Function Approximation** Martens et al. [2010] first approximated the hessian of deep networks using the conjugate gradient method, and Koh and Liang [2017] applied this to influence functions. Kwon et al. [2023] and Grosse et al. [2023] propose algorithms that efficiently approximate the conjugate gradient method, and apply them to LoRA-finetuned and pretrained language models, respectively. They do not evaluate the conjugate gradient method for data curation, nor do they consider the task of preference modeling.

**RLHF and Data Curation** Stiennon et al. [2020] showed that a language model could be taught to generate better summaries using human feedback in the form of pairwise preferences. We use the TL;DR dataset they provide. Bai et al. [2022a] post-trained a language model to align it's behavior to be helpful and harmless. Morimura et al. [2024] and Gao et al. [2024] point out and attempt to address low data quality for preference data. The former does this by proposing a framework for data collection, the latter by proposing filtering based on trained reward-model logits.

## 3 Methods

### 3.1 Task and Data

We study the human–preference reward–modeling task introduced by Bai et al. [2022b] on the public TL;DR dataset released by OpenAI.[1] Each training example is a chosen and a rejected summary of the same news article, labeled by crowd workers. Following common practice, we cast the task as a Bradley–Terry pairwise–ranking problem: the reward model $f_\theta$ should assign a higher scalar reward to the chosen summary than to its rejected counterpart.

**Dataset modification**. In order to make the problem tractable under a very tight computation budget, we filter out all examples for which both summaries are longer than 24 tokens Llama3 tokens, and discard the original post text, using only summary preferences and summaries themselves. This leaves us with roughly 8.5k examples total. For influence estimation we sample $|\mathcal{D}_{\text{val}}| = 100$ validation pairs uniformly at random from the held-out validation set. All remaining validation data serve as an untouched test set for final evaluation.

### 3.2 Base Model and Fine-tuning Setup

Our base model is LLaMA-3.2-1B. We apply Low-Rank Adaptation (LoRA; Hu et al., 2022) with rank $r=8$, $\alpha=16$ and dropout 0.05 to the projection matrices $\{q, k, v, o\}$. Fine-tuning uses the AdamW optimiser ($\beta_1=0.9$, $\beta_2=0.98$, $\epsilon=10^{-8}$) for three epochs, batch size 124 and learning rate $1 \times 10^{-5}$ with cosine decay. Only LoRA parameters ($\approx 0.12\%$ of total weights) are updated.

### 3.3 Influence–Function Approximation

The goal is to estimate, for every training example $z_i$ and validation example $z_j$, the classical influence value $\mathcal{I}_{\text{IF}}(z_i, z_j) = -\nabla_\theta L(z_j)^\top H_\theta^{-1} \nabla_\theta L(z_i)$, where $H_\theta$ is the Hessian of the total training loss. Instead of inverting a billion-dimensional Hessian, we study only the LoRA adapter weights. These account for 0.12% of all parameters yet dominate post-training behavior.

Computing $H_\theta^{-1} \nabla_\theta L(z_j)$ is equivalent to solving the linear system $(H_\theta + \lambda I) x = g_j$ with $g_j := \nabla_\theta L(z_j)$, where we add Tikhonov damping $\lambda=10^{-2}$ to guarantee positive definiteness. This linear system is the first-order optimality condition of the strictly convex quadratic $\min_{x \in \mathbb{R}^d} \frac{1}{2} x^\top (H_\theta + \lambda I)x - g_j^\top x$. To computing an HVP without forming $H_\theta$, we use the double-back-prop trick [Pearlmutter, 1994]. We average the operation over a mini-batch of $|\mathcal{B}| = 20$ training examples.

### 3.4 Evaluation

Let $\bar{\mathcal{I}}(z_i) = \frac{1}{|\mathcal{D}_{\text{val}}|} \sum_{z_j \in \mathcal{D}_{\text{val}}} \mathcal{I}_{\text{IF}}(z_i, z_j)$ denote the mean influence of a training example across validation points. We rank all $z_i \in \mathcal{D}_{\text{train}}$ by $\bar{\mathcal{I}}(z_i)$ and drop the worst $x\%$ examples. After pruning,

---

[1] https://github.com/openai/summarize-from-feedback

**Algorithm 1** Influence computation in the LoRA parameter space

**Require:** LoRA-tuned model $f_\theta$; training set $\mathcal{D}_{\text{train}}$; validation set $\mathcal{D}_{\text{val}}$; damping $\lambda$; batch size $B$; CG iteration budget $K$

1: **for all** $z_j \in \mathcal{D}_{\text{val}}$ **do**
2:      $g_j \leftarrow \nabla_\theta L(z_j)$
3:      $x_j \leftarrow 0; r \leftarrow g_j; p \leftarrow r$
4:      **for** $k = 1$ to $K$ **do**
5:          $\mathcal{B} \leftarrow$ sample $B$ elements from $\mathcal{D}_{\text{train}}$
6:          $h \leftarrow \frac{1}{|\mathcal{B}|} \sum_{z_i \in \mathcal{B}} \nabla_\theta^2 L(z_i)\, p + \lambda p$
7:          $\alpha \leftarrow \frac{\langle r, r \rangle}{\langle p, h \rangle}$
8:          $x_j \leftarrow x_j + \alpha p$
9:          $r_{\text{new}} \leftarrow r - \alpha h$
10:         **if** $\|r_{\text{new}}\| < \varepsilon$ **then**
11:            break
12:         **end if**
13:         $\beta \leftarrow \frac{\langle r_{\text{new}}, r_{\text{new}} \rangle}{\langle r, r \rangle}$
14:         $p \leftarrow r_{\text{new}} + \beta p$
15:         $r \leftarrow r_{\text{new}}$
16:      **end for**
17:      $x_j \leftarrow \textbf{detach}(x_j)$
18:      **for all** $z_i \in \textbf{local\_shard}(\mathcal{D}_{\text{train}})$ **do**
19:          $g_i \leftarrow \nabla_\theta L(z_i)$
20:          $\mathcal{I}(z_i, z_j) \leftarrow -\langle x_j, g_i \rangle$
21:      **end for**
22: **end for**
23: **return** $\mathcal{I}$ collected on rank 0

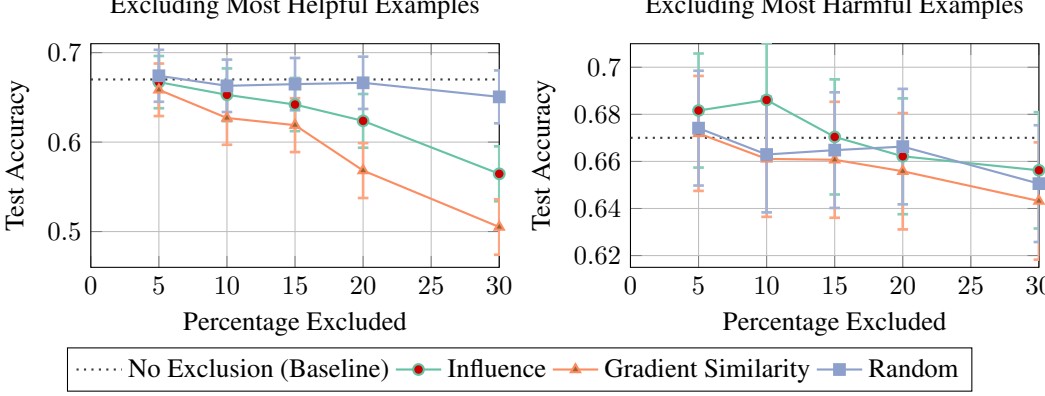

Figure 1: Test accuracy after excluding training examples. Left: removing the most helpful examples. Right: removing the most harmful ones. Vertical bars show 95% Wald Interval.

we retrain the reward model from the original checkpoint under the identical hyper-parameter schedule. We report pairwise accuracy on the untouched test split. We use two baselines. The first is random removal of $x\%$ of training examples. The second is gradient similarity, a first-order influence approximation that presumes the Hessian to be the identity Dhaliwal and Shintre [2018].

## 4 Results

Figure 1 shows that influence-based filtering improves performance by 1.5% above finetuning on the entire dataset when 10% of examples are pruned. Accuracy gains are not statistically significant, but influence-pruning does outperform other pruning techniques at 10% exclusion. Gradient similarity performs similarly to random pruning for harmful examples.

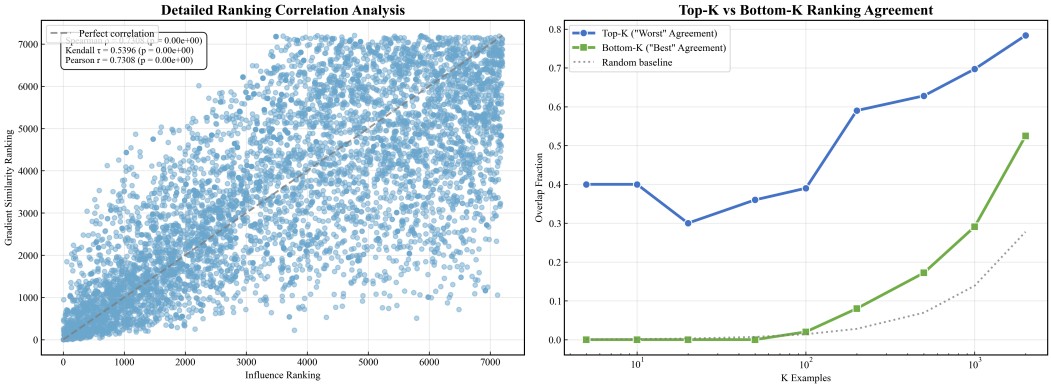

Figure 2: Left: Rank correlation between Influence and Gradient Similarity Rankings (lower ranks represent worse examples). Right: Ranking agreement between gradient similarity and influence function approaches for top/bottom k examples.

Interestingly, gradient similarity outperforms influence function approximation for finding the most helpful examples. Removing 30% of the best training examples identified by gradient similarity reduces performance of the trained reward model to random chance, while influence-based pruning only reduces performance to roughly 56%. Figure 2 shows that there is strong agreement among what are deemed to be the worst examples, but only near random chance agreement in what is deemed to be the best examples.

## 5 Discussion

We hypothesize that the failure of gradient similarity on finding harmful examples has to do with the more pronounced local curvature around these harmful examples. Helpful examples are those which give the model new information about a particular context. This corresponds to relatively a flat part of parameter space where not much has been learned, and therefore gradient similarity and influence functions both work well. Harmful examples, on the other hand, are likely those which sit on steep inclines in the direction that go against the consensus found in the training data. Though the gradient similarity approach can determine when a given training data is poorly suited for a given validation example, it cannot determine the extent to which this error is surprising, or merely noise. This finding suggests further work that might be done to more efficiently find harmful examples to advance data curation at scale.

## 6 Conclusion

Influence function approximations offer a promising approach to filtering noisy human preference data for reward model training. First-degree approximation via gradient similarity appears more effective at finding helpful training examples, but less effective at finding harmful ones. Future work may attempt to use the insight that training data that disagrees with validation data in a way that is surprising are most likely to be harming test performance to devise more efficient methods of data curation.

## 7 Limitations

Calculating influence function approximations is computationally expensive, even at smaller scales. Thus, we are constrained to using a single small dataset. Datasets are often generated in vastly different ways, and likely have highly variable noise. This limits the generalization of these results beyond the context of crowd-sourced human preference data.

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

# A  Compute Requirements

We used an internal node containing 4 L40 GPUs for 20 hours total. Most of the compute went into calculating influence function approximations and LoRA retraining.

