# OpenReview forum: "Influence Functions for Preference Dataset Pruning"
_NeurIPS.cc/2025/Workshop/Reliable_ML — NeurIPS 2025 - Reliable ML Workshop_

### Official Review · Reviewer_ie3m · 2025-09-08
**Interesting method, but might be out of application range of influence function when dropping 30% of data**

**Rating:** 6
**Confidence:** 3

**Review:**

The author applied influence function approximation to evaluate data pruning in a reward model setup. This is a sensible method but interestingly performs worse than gradient similarity. I suspect this is because dropping proportion is >5% while influence function is first order Taylor expansion and 5-30% might be out of its application range.

---

### Official Review · Reviewer_eUQZ · 2025-09-19
**CG influence functions slightly improve preference-data pruning with broader evidence needed.**

**Rating:** 6
**Confidence:** 2

**Review:**

This paper applies conjugate-gradient influence functions in the LoRA parameter space of LLaMA-3.2-1B to prune noisy TL;DR preference pairs and retrain under identical hyperparams. Pruning the 10% most harmful examples yields a ~+1.5% test-accuracy uplift, with 95% Wald CIs reported (effects are modest).

Pros: clear setup & parity retraining; sensible HVP/CG implementation; honest limitations.

Cons: effect size is small; single dataset/model; compute trade-off vs. gain not quantified.

I suggest adding another preference dataset/model size, reporting wall-clock/GPU hours per % gain, comparing to TracIn/Data-Shapley and including qualitative analysis of pruned items.